# EFFICIENT TRANSFER LEARNING IN DIFFUSION MODELS VIA ADVERSARIAL NOISE

## ABSTRACT

Diffusion Probabilistic Models (DPMs) have demonstrated substantial promise in image generation tasks but heavily rely on the availability of large amounts of training data. Previous works, like GANs, have tackled the limited data problem by transferring pre-trained models learned with sufficient data. However, those methods are hard to be utilized in DPMs since the distinct differences between DPM-based and GAN-based methods, showing in the unique iterative denoising process integral and the need for many timesteps with no-targeted noise in DPMs. In this paper, we propose a novel DPMs-based transfer learning method, TAN, to address the limited data problem. It includes two strategies: similarity-guided training, which boosts transfer with a classifier, and adversarial noise selection which adaptively chooses targeted noise based on the input image. Extensive experiments in the context of few-shot image generation tasks demonstrate that our method is not only efficient but also excels in terms of image quality and diversity when compared to existing GAN-based and DDPM-based methods.

## 1 INTRODUCTION

Generative models, such as GANs (Brock et al., 2018; Khan et al., 2022), VAEs (Kingma & Welling, 2013; Rezende et al., 2014), and autoregressive models (Van den Oord et al., 2016; Chen et al., 2018; Grill et al., 2020), have made remarkable successes in various fields across images (Brock et al., 2018; Razavi et al., 2019), text (Brown et al., 2020), and audio (Dhariwal et al., 2020; Oord et al., 2016) by utilizing vast amounts of unlabeled data for training. Diffusion probabilistic models (DPMs) (Sohl-Dickstein et al., 2015; Ho et al., 2020; Nichol & Dhariwal, 2021), which are designed to replicate data distributions by learning to invert multi-step noise procedures, have recently experienced significant advancements, enabling the generation of high-definition images with broad diversity. Although DPMs have emerged as a potent tool for image generation with remarkable results in terms of both quality and diversity, modern DPMs heavily rely on extensive amounts of data to train the large-scale parameters of their networks (Cao et al., 2022). This dependency can lead to overfitting and a failure to generate diverse, high-quality images with limited training data. Unfortunately, gathering sufficient data is not always feasible in certain situations.

Transfer learning can be an effective solution to this challenge, as it applies knowledge from a pre-trained generative model trained on a large dataset to a smaller one. The fundamental idea is to begin training with a source model that has been pre-trained on a large dataset, and then adapt it to a target domain with limited data. Several techniques have been proposed in the past to adapt pre-trained GAN-based models (Wang et al., 2018; Karras et al., 2020a; Wang et al., 2020; Li et al., 2020) from large-scale source datasets to target datasets using a limited number of training samples. Typically, methods for few-shot image generation either enhance the training data artificially using data augmentation to prevent overfitting (Zhang et al., 2018; Karras et al., 2020a), or directly evaluate the distance between the processed image and the target image (Ojha et al., 2021; Zhao et al., 2022).

Nevertheless, applying prior GAN-based techniques to DPMs is challenging due to the differences in training processes between GAN-based and DPM-based methods. GANs can quickly generate a final processed image from latent space, while DPMs only predict less noisy images at each step and request large timesteps to generate a high-quality final image. Such an iterative denoising process poses two challenges when transferring diffusion models. The first challenge is that the transfer direction needs to be estimated on noisy images. The single-pass generation of GANs allows them

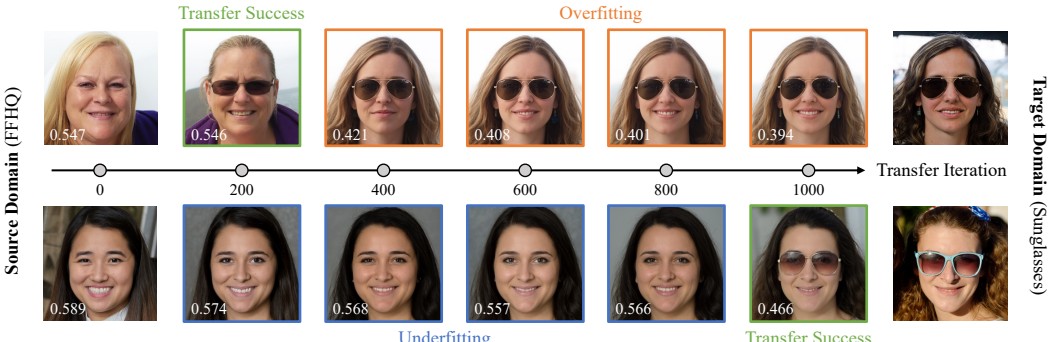

Figure 1: Two sets of images generated from corresponding fixed noise inputs at different stages of fine-tuning DDPM from FFHQ to 10-shot Sunglasses. The perceptual distance (LPIPS Zhang et al. (2018)) with the training target image is shown on each generated image. When the bottom image successfully transfers to the target domain, the top image has already suffered from overfitting.

to directly compare the generated clean images with the target image (Li et al., 2020; Ojha et al., 2021; Zhao et al., 2022), which is not easily applicable to diffusion models. The current DPM-based few-shot method, DDPM pairwise adaptation (DDPM-PA) (Zhu et al., 2022), substitutes the high-quality real final image with the predicted blurry final at the intermediate timestep to address this issue. However, comparing the target image with the blurry image can be problematic and inaccurate, as the predicted image may not accurately represent the domain of generated images. It leads to the production of DDPM-PA final images that are fuzzy and distorted. Moreover, even if the transfer direction can be available, we still face a more fundamental second challenge resulting from the noise mechanism in diffusion models. The diffusion and denoising process utilize fully random Gaussian noise, which is independent of the input image and makes no assumption of it. We observe that such non-targeted noise imposes unbalanced effects on different images, leading to divergent transferring pace in terms of training iteration needed. As demonstrated in Figure 1, when one image (below) is just successfully transferred from the source domain to the target domain, another image (above) may have severely overfit and become too similar to the target image. Such normally distributed noise may also necessitate an extensive number of iterations to transfer, especially when the gradient direction is noisy due to limited images.

In this paper, to handle the challenge of transferring direction estimation for diffusion models, we propose to leverage a similarity measurement to estimate the gap between the source and the target, which circumvents the necessity of comparing individual images. Building upon this, we introduce a ***similarity-guided training*** approach to fine-tune the pre-trained source model to the target domain. It employs a classifier to estimate the divergence between the source and target domains, leveraging existing knowledge from the source domain to aid in training the target domain. This method not only helps in bridging the gap between the source and target domains for diffusion models but also addresses the unstable gradient direction caused by limited target data in the few-shot setting by implicitly comparing the sparse target data with the abundant source data. More importantly, to tackle the challenge of non-targeted noise in diffusion models, we propose a novel min-max training process, i.e., ***adversarial noise selection***, to dynamically choose the noise according to the input image. The adversarial noise scheme enhances few-shot transfer learning by minimizing the "worse-case" Gaussian noise which the pre-trained model fails to denoise on the target dataset. This strategy also significantly reduces the training iterations needed and largely improves the efficiency of the transfer learning for diffusion models. Our adversarial strategy with similarity measurement excels in few-shot image generation tasks, speeding up training, achieving quicker convergence, and creating images fitting the target style while resembling source images. Our Experiments on few-shot image generation tasks demonstrate our method surpasses existing GAN-based and DDPM-based techniques, offering superior quality and diversity.

## 2 RELATED WORK

### 2.1 DIFFUSION PROBABILISTIC MODELS

Ho et al. (2020) has been leveraged as an effective generative model that circumvents the adversarial training inherent in GANs (Goodfellow et al., 2020). DDPMs, by enabling the diffusion reverse process, are capable of reconstructing images. However, due to their extensive iterative time steps,

DDPMs are subject to the challenge of high computational time. DDIM (Song et al., 2020) addresses this issue by "implicating" the model, which allows it to function with far fewer iterations and dramatically reduces the inference time compared to DDPM. Conversely, a fresh approach to the diffusion model is the score-based model via SDE, wherein the diffusion and the denoising processes are both modeled by SDEs. Song & Ermon (2019) initially proposed the generation of samples from latent noise via the Dynamic Langevin Sampling method. For fast high-quality, high-resolution image generation, Latent Diffusion Models (LDMs) (Rombach et al., 2022) propose an advanced machine learning methodology by gradually transforming a random noise into the target image through a diffusion process that uses latent space representations.

## 2.2 FEW-SHOT IMAGE GENERATION

Existing practices predominantly adopt an adaptation pipeline where a foundational model is pre-trained on a large source domain, and then adjusted to a smaller target domain. In contrast, few-shot image generation strives to envision new and diverse examples while circumventing overfitting to the limited training images. FreezeD (Mo et al., 2020) addresses overfitting by locking parameters in the high-resolution layers of the discriminator. EWC (Li et al., 2020) utilizes elastic weight consolidation, making it difficult to modify essential weights that possess high Fisher information values. CDC (Ojha et al., 2021) introduces a cross-domain consistency loss and patch-level discrimination to forge a connection between the source and target domains. DCL (Zhao et al., 2022) uses contrastive learning to distance the generated samples from actual images and maximize the similarity between corresponding image pairs in source and target domains. The DDPM-PA (Zhu et al., 2022) adopts a similar approach to CDC for adapting models pre-trained on extensive source domains to target domains. GAN-based methods, like CDC and DCL, require the final generated image during training. In contrast, DPMs' training process aims to predict the next stage of noised images and can only yield a blurry predicted image during the training stage.

## 3 PRELIMINARY

Gaussian diffusion models are used to approximate the data distribution $x_0 \sim q(x_0)$ by $p_\theta(x_0)$. The distribution $p_\theta(x_0)$ is modeled in the form of latent variable models. According to (Ho et al., 2020), the diffusion process from a data distribution to a Gaussian distribution with variance $\beta_t \in (0,1)$ for timestep $t$ can be expressed as:

$$q(x_t|x_0) = \mathcal{N}(x_t; \bar{\alpha}_t x_0, (1-\bar{\alpha}_t)\mathbf{I}),$$
$$x_t = \sqrt{\bar{\alpha}_t}x_0 + \sqrt{1-\bar{\alpha}_t}\epsilon,$$

where $\alpha_t := 1 - \beta_t$, $\bar{\alpha}_t := \prod_{i=0}^{t}(1-\beta_i)$ and $\epsilon \sim \mathcal{N}(\mathbf{0},\mathbf{I})$. Ho et al. (2020) trains a U-Net (Ronneberger et al., 2015) model parameterized by $\theta$ to fit the data distribution $q(x_0)$ by maximizing the variational lower-bound. The DDPM training loss with model $\epsilon_\theta(x_t,t)$ can be expressed as:

$$\mathcal{L}_{\texttt{sample}}(\theta) := \mathbb{E}_{t,x_0,\epsilon}\left\|\epsilon - \epsilon_\theta(x_t,t)\right\|^2. \tag{1}$$

Based on (Song et al., 2020), the reverse process of DDPM and DDIM at timestep $t$ can be expressed as:

$$x_{t-1} = \sqrt{\bar{\alpha}_{t-1}}\underbrace{\left(\frac{x_t - \sqrt{1-\bar{\alpha}_t}\epsilon_\theta(x_t,t)}{\bar{\alpha}_t}\right)}_{\text{predicted } \mathbf{x_0}} + \underbrace{\sqrt{1-\bar{\alpha}_{t-1}-\sigma_t^2}\cdot\epsilon_\theta(x_t,t)}_{\text{direction pointing to } \mathbf{x_t}} + \underbrace{\sigma_t\epsilon_t}_{\text{random noise}},$$

where $\sigma_t = \eta\sqrt{(1-\bar{\alpha}_{t-1})/(1-\bar{\alpha}_t)}\sqrt{1-\bar{\alpha}_t/\bar{\alpha}_{t-1}}$ and $\eta = 0$ (Song et al., 2020) or $\eta = 1$ (Ho et al., 2020) or $\eta = \sqrt{(1-\bar{\alpha}_t)/(1-\bar{\alpha}_{t-1})}$ (Ho et al., 2020). Enhance, Dhariwal & Nichol (2021) propose the conditional reverse noise process as:

$$p_{\theta,\phi}(x_{t-1}|x_t,y) \approx \mathcal{N}(x_{t-1}; \mu_\theta(x_t,t) + \sigma_t^2\gamma\nabla_{x_t}\log p_\phi(y|x_t), \sigma_t^2\mathbf{I}), \tag{2}$$

$$\text{where}\quad \mu_\theta(x_t,t) = \frac{1}{\sqrt{\alpha_t}}\left(x_t - \frac{1-\alpha_t}{\sqrt{1-\bar{\alpha}_t}}\epsilon_\theta(x_t,t)\right), \tag{3}$$

and $\gamma$ is a hyperparameter for conditional control. For the sake of clarity in distinguishing domains, this paper uses $\mathcal{S}$ and $\mathcal{T}$ to represent the source and target domain, respectively.

## 4    TRANSFER LEARNING IN DIFFUSION MODELS VIA ADVERSARIAL NOISE

In this section, we introduce transfer learning in diffusion models via Adversarial Noise, dubbed TAN, with similarity-guided training and adversarial noise selection for stronger transfer ability.

### 4.1    SIMILARITY-GUIDED TRAINING

We use similarity to measure the gap between the source and target domains using a noised image $x_t$ at timestep $t$ instead of the final image. Drawing inspiration from (Dhariwal & Nichol, 2021; Liu et al., 2023), we express the domain difference between the source and target in terms of the divergence in similarity measures. Initially, we assume a model that can predict noise with both source and target domains, denoted as $\theta_{(\mathcal{S},\mathcal{T})}$. As Equation 2, the reverse process for the source and target images can be written as:

$$p_{\theta_{(\mathcal{S},\mathcal{T})},\phi}(x_{t-1}|x_t, y = Y) \approx \mathcal{N}(x_{t-1}; \mu_{\theta_{(\mathcal{S},\mathcal{T})}} + \sigma_t^2 \gamma \nabla_{x_t} \log p_\phi(y = Y|x_t), \sigma_t^2 \mathbf{I}) , \qquad (4)$$

where $Y$ is $\mathcal{S}$ or $\mathcal{T}$ for source or target domain image generation, respectively. We can consider $\mu(x_t) + \sigma_t^2 \gamma \nabla_{x_t} \log p_\phi(y = \mathcal{S}|x_t)$ as the source model $\theta_{\mathcal{S}}$, which only synthesize image on the source domain respectively. For brevity, we denote $p_{\theta_{\mathcal{S}},\phi}(x_{t-1}^{\mathcal{S}}|x_t) = p_{\theta_{(\mathcal{S},\mathcal{T})},\phi}(x_{t-1}|x_t, y = \mathcal{S})$. We similar define $p_{\theta_{\mathcal{T}},\phi}(x_{t-1}^{\mathcal{T}}|x_t)$ as above by replace $\mathcal{S}$ with $\mathcal{T}$. Therefore, the KL-divergence between the output of source model $\theta_{\mathcal{S}}$ and the target $\theta_{\mathcal{T}}$ with the same input $x_t$ at timestep $t$, is defined as:

$$\mathrm{D_{KL}} \left( p_{\theta_{\mathcal{S}},\phi}(x_{t-1}^{\mathcal{S}}|x_t), p_{\theta_{\mathcal{T}},\phi}(x_{t-1}^{\mathcal{T}}|x_t) \right)$$
$$= \mathbb{E}_{t,x_0,\epsilon} \left[ \|\nabla_{x_t} \log p_\phi(y = \mathcal{S}|x_t) - \nabla_{x_t} \log p_\phi(y = \mathcal{T}|x_t)\|^2 \right] , \qquad (5)$$

where $p_\phi$ is a classifier to distinguish $x_t$. The detailed derivation is in Appendix. We consider the $\nabla_{x_t} \log p_\phi(y = \mathcal{S}|x_t)$ and $\nabla_{x_t} \log p_\phi(y = \mathcal{T}|x_t)$ as the similarity measures of the given $x_t$ in the source and target domains, respectively. Since transfer learning primarily focuses on bridging the gap between the image generated by the current fine-tuning model and the target domain image, we disregard the first term and utilize only $p_\phi(y = \mathcal{T}|x_t^{\mathcal{T}})$ to guide the training process. Specifically, we employ a fixed pre-trained binary classifier that differentiates between source and target images at time step $t$ to boost the training process. Similarly with the vanilla training loss in DPMs (Ho et al., 2020), i.e., Equation 1, we use the KL-divergence between the output of current model $\theta$ and target model $\theta_{\mathcal{T}}$ at time step $t$ as:

$$\min_\theta \mathbb{E}_{t,x_0,\epsilon} \left[ \left\| \epsilon_t - \epsilon_\theta(x_t, t) - \hat{\sigma}_t^2 \gamma \nabla_{x_t} \log p_\phi(y = \mathcal{T}|x_t) \right\|^2 \right] , \qquad (6)$$

where $\epsilon_t \sim \mathcal{N}(\mathbf{0}, \mathbf{I})$, $\epsilon_\theta$ is the pre-trained neural network on source domain, $\gamma$ is a hyper-parameter to control the similarity guidance, $\hat{\sigma}_t = (1 - \bar{\alpha}_{t-1}) \sqrt{\frac{\alpha_t}{1 - \bar{\alpha}_t}}$, and $p_\phi$ is the binary classifier differentiating between source and target images. Equation 6 is defended as similarity-guided DPMs train loss. The full proof is provided in the Appendix. We leverage the pre-trained classifier to indirectly compare the noised image $x_t$ with both domain images, subtly expressing the gap between the currently generated image and the target image. By minimizing the output of the neural network with corrected noise, we bridge the gap in the diffusion model and bolster transfer learning. Furthermore, similarity guidance enhances few-shot transfer learning by avoiding misdirection towards the target image, as $\nabla_{x_t} \log p_\phi(y = \mathcal{T}|x_t)$ acts as an indirect indicator, rather than straightly relying on the original image.

### 4.2    ADVERSARIAL NOISE SELECTION

Despite potentially determining the transfer direction, we still encounter a fundamental second challenge originating from the noise mechanism in diffusion models. As mentioned, the model needs to be trained to accommodate the quantity of noise $\epsilon_t$ over many iterations. However, increasing iterations with limited images may lead to overfitting of the training samples, thereby reducing the diversity of the generated samples. On the other hand, training with too few iterations might only successfully transform a fraction of the generated images into the target domain as Figure 1.

To counter these issues, we propose an adaptive noise selection method. This approach utilizes a min-max training process to reduce the actual training iterations required and ensure the generated images closely resemble the target images. After the model has been trained on a large dataset, it

exhibits a strong noise reduction capability for source datasets. This implies it only needs to minimize specific types of Gaussian noise with which the trained model struggles or fails to denoise with the target domain sample. The first step in this process is to identify the maximum Gaussian noise with the current model, and then specifically minimize the model using this noise. Based on Equation 6, this can be mathematically formulated as follows:

$$\min_{\theta} \max_{\epsilon} \mathbb{E}_{t,x_0} \left[ \left\| \epsilon - \epsilon_{\theta}(x_t, t) - \hat{\sigma}_t^2 \gamma \nabla_{x_t} \log p_{\phi}(y = \mathcal{T}|x_t) \right\|^2 \right] . \tag{7}$$

Although finding the exact maximum noise is challenging as Equation 7, the projected gradient descent (PGD) strategy can be used to solve the inner maximization problem instead. Specifically, the inner maximization of Gaussian noise can be interpreted as finding the "worse-case" noise corresponding to the current neural network. Practically, the similarity-guided term is disregarded, as this term is hard to compute differential and is almost unchanged in the process. We utilize the multi-step variant of PGD with gradient ascent, as expressed below:

$$\epsilon^{j+1} = \text{Norm} \left( \epsilon^j + \omega \nabla_{\epsilon^j} \left\| \epsilon^j - \epsilon_{\theta}(\sqrt{\bar{\alpha}_t} x_0 + \sqrt{1 - \bar{\alpha}_t} \epsilon^j, t) \right\|^2 \right), \quad j = 0, \cdots, J - 1, \tag{8}$$

where $\omega$ is a hyperparameter that represents the "learning rate" of the negative loss function, and Norm is a normalization function that approximately ensures the mean and standard deviation of $\epsilon^{j+1}$ is $\mathbf{0}$ and $\mathbf{I}$ respectively. The initial value, $\epsilon_0$, is sampled from the Gaussian distribution as $\epsilon_0 \sim \mathcal{N}(\mathbf{0}, \mathbf{I})$. We use this method to identify this worse-case noise and minimizing the "worse-case" Gaussian noise is akin to minimizing all Gaussian noises that are "better" than it. By adaptively choosing this specific noise, we can more accurately correct the gradient to enhance training with limited data, effectively addressing the underfitting problem during a limited number of iterations.

### 4.3 OPTIMIZATION

For time and GPU memory saving, we implement an additional adaptor module, $\psi^l$, (Noguchi & Harada, 2019) to learn the shift gap as Equation 5 based on $x_t$ in practice. During the training, we keep the parameters of $\theta^l$ constant and update the additional adaptor layer parameters $\psi^l$. The overall loss function can be expressed as follows,

$$L(\psi) \equiv \mathbb{E}_{t,x_0} \left\| \epsilon^{\star} - \epsilon_{\theta,\psi}(x_t^{\star}, t) - \hat{\sigma}_t^2 \gamma \nabla_{x_t^{\star}} \log p_{\phi}(y = \mathcal{T}|x_t^{\star}) \right\|^2 \tag{9}$$

$$\text{s.t. } \epsilon^{\star} = \arg\max_{\epsilon} \left\| \epsilon - \epsilon_{\theta}(\sqrt{\bar{\alpha}_t} x_0 + \sqrt{1 - \bar{\alpha}_t} \epsilon, t) \right\|^2 , \quad \epsilon_{\text{mean}}^{\star} = \mathbf{0} \text{ and } \epsilon_{\text{std}}^{\star} = \mathbf{I} , \tag{10}$$

where $\epsilon^{\star}$ is the "worse-case" noise, the $x_t^{\star} = \sqrt{\bar{\alpha}_t} x_0 + \sqrt{1 - \bar{\alpha}_t} \epsilon^{\star}$ is the corresponding noised image at the timestep $t$ and $\psi$ is certain extra parameter beyond pre-trained model. We link the pre-trained U-Net model with the adaptor layer (Houlsby et al., 2019) as $x_t^l = \theta^l(x_t^{l-1}) + \psi^l(x_t^{l-1})$, where $x_t^{l-1}$ and $x_t^l$ represents the $l$-th layer of the input and output, and $\theta^l$ and $\psi^l$ denote the $l$-th layer of the pre-trained U-Net and the additional adaptor layer, respectively.

---

**Algorithm 1** Training DPMs with TAN

---

**Require:** binary classifier $p_{\phi}$, pre-trained DPMs $\epsilon_{\theta}$, learning rate $\eta$
1: **repeat**
2:     $x_0 \sim q(x_0)$;
3:     $t \sim \text{Uniform}(\{1, \cdots, T\})$;
4:     $\epsilon \sim \mathcal{N}(\mathbf{0}, \mathbf{I})$;
5:     **for** $j = 0, \cdots, J - 1$ **do**
6:         Update $\epsilon^j$ via Eq. 8;
7:     **end for**
8:     Compute $L(\psi)$ with $\epsilon^{\star} = \epsilon^J$ via Eq. 9;
9:     Update the adaptor model parameter: $\psi = \psi - \eta \nabla_{\psi} L(\psi)$;
10: **until** converged.

---

The full training procedure of our method, named DPMs-TAN, is outlined in Algorithm 1. Initially, as in the traditional DDPM training process, we select samples from target datasets and randomly choose a timestep $t$ and standard Gaussian noise for each sample. We employ limited extra adaptor

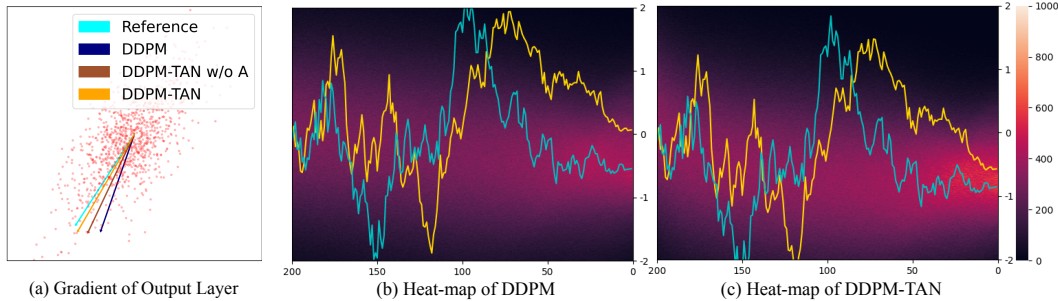

(a) Gradient of Output Layer  (b) Heat-map of DDPM  (c) Heat-map of DDPM-TAN

Figure 2: This Figure visualizes gradient changes and heat maps: Figure (a) shows gradient directions with various settings—the cyan line for the gradient of 10,000 samples in one step, dark blue for ten samples in one step as baseline method (trained with traditional DDPM), the sienna for our similarity-guided training, and the orange for our method DDPM-TAN, while red points at the background are "worse"-case noises by adversarial noise selection; Figure (b) and (c) depict heat-maps of the baseline and our method, with cyan and gold lines representing the generation sampling process value with the original DDPM and our method, respectively.

module parameters with the pre-train model. Subsequently, we identify the adaptive inner max as represented in Equation 8 with the current neural network. Based on these noises, we compute the similarity-guided DDPM loss as Equation 6, which bridges the discrepancy between the pre-trained model and the scarce target samples. Lastly, we execute backpropagation to only update the adaptor module parameters.

## 5 EXPERIMENT

To demonstrate the effectiveness of our approach, we perform a series of few-shot image generation experiments using a limited set of just 10 training images with the same setting as DDPM-PA (Zhu et al., 2022). We compare our method against state-of-the-art GAN-based and DDPM-based techniques, assessing the quality and diversity of the generated images through both qualitative and quantitative evaluations. This comprehensive comparison provided strong evidence of the superiority of our proposed method in the context of few-shot image generation tasks.

### 5.1 VISUALIZATION ON TOY DATA

To conduct a quantitative analysis, we trained a diffusion model to generate 2-dimensional toy data with two Gaussian noise distributions. The means of the Gaussian noise distributions for the source and target are $(1, 1)$ and $(-1, -1)$, and their variances are denoted by $\mathbf{I}$. We train a simple neural network with source domain samples and then transfer this pre-trained model to target samples.

Figure 2 (a) illustrates the output layer gradient direction of four different settings in the first iteration, with the same noise and timestep $t$. The red line, computed with ten thousand different samples, is considered a reliable reference direction (close to 45 degrees southwest). For 10-shot samples, we repeat them a thousand times into one batch to provide a unified comparison with ten thousand different samples. The dark blue line and the sienna represent the gradient computed with the traditional DDPM as the baseline and similarity-guided training in a 10-shot sample, respectively. The orange line represents our method, DDPM-TAN, in a 10-shot sample. The gradient of our method is closer to the reliable reference direction, demonstrating that our approach can effectively correct the issue of the noisy gradient. The red points in the background symbolize "worse-case" noise, which is generated through adversarial noise selection. The graphic shows how the noise distribution transitions from a circle (representing a normal Gaussian distribution) to an ellipse. The principal axis of this ellipse is oriented along the gradient of the model parameters. This illustrates the noise distribution shift under our adversarial noise selection approach, which effectively fine-tunes the model by actively targeting the "worst-case" noise that intensifies the adaptation task.

Figures 2 (b) and (c) present heatmaps of the baseline and our method in only one dimension, respectively. The cyan and gold lines denote the values of the generation sampling process using the original DDPM and our method. The heat-maps in the background illustrate the distribution of values for 20,000 samples generated by the original DDPM (baseline) and our method. The lighter the color in the background, the greater the number of samples present. There is a significantly brighter

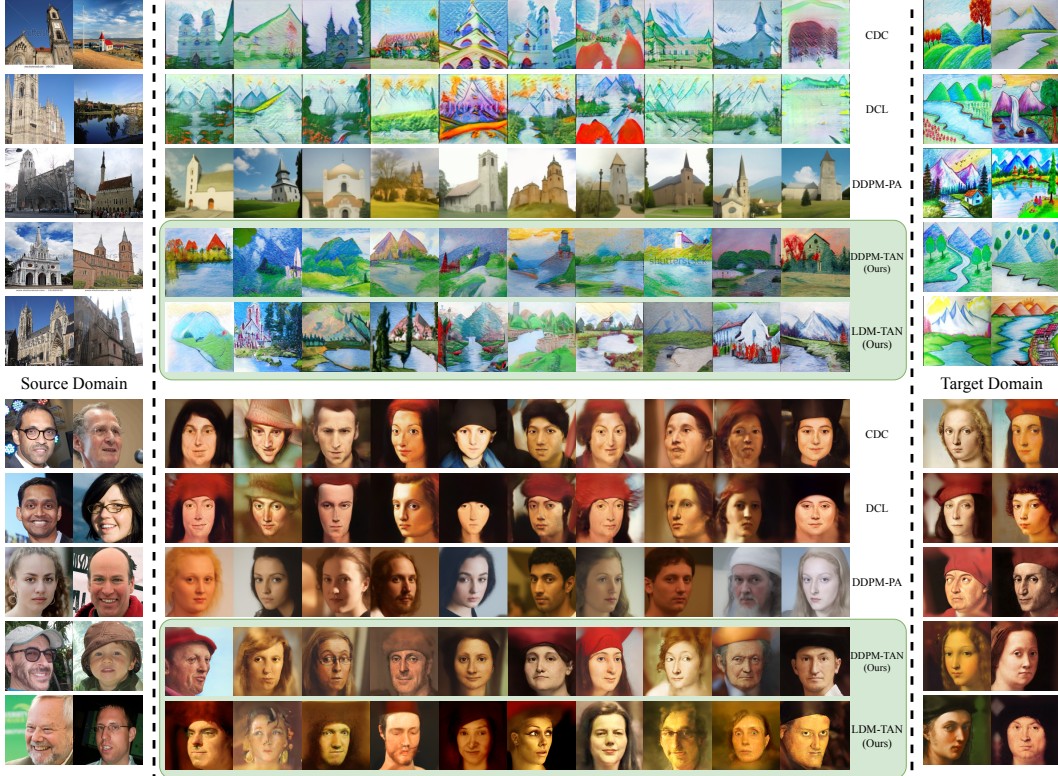

Figure 3: The 10-shot image generation samples on LSUN Church → Landscape drawings (top) and FFHQ → Raphael's paintings (bottom). When compared with other GAN-based and DDPM-based methods, our method, TAN, yields high-quality results that more closely resemble images of the target domain style, with less blurring.

central highlight in (c) compared to (b), demonstrating that our method can learn the distribution more quickly than the baseline method. The gold and cyan lines in the two figures are approximately parallel, providing further evidence that our method can learn the gap more rapidly.

## 5.2 EXPERIMENTAL SETUP

**Datasets.** Following (Ojha et al., 2021), we use FFHQ (Karras et al., 2020b) and LSUN Church (Yu et al., 2015) as source datasets. For the target datasets, we employ 10-shot Sketches, Babies, Sunglasses, and face paintings by Amedeo Modigliani and Raphael Peale, which correspond to the source domain FFHQ. Additionally, we utilize 10-shot Haunted Houses and Landscape drawings as target datasets corresponding to the LSUN Church source domain.

**Configurations.** We evaluate our method not only on the DDPM framework but also on the LDMs. For this, we employ a pre-trained DDPM similar to DDPM-PA and use the pre-trained LDMs as provided in (Rombach et al., 2022). We restrict our fine-tuning to the shift module of the U-Net, maintaining the pre-trained DPMs and autoencoders in LDMs as they are. For similarity-guided training, we establish $\gamma = 5$ and we utilized a model pre-trained on the ImageNet dataset and subsequently fine-tuned it with using a new classifier head on a limited set of 10 target domain images. In the case of adversarial noise selection, we assign $J = 10$ and $\omega = 0.02$ for most transfer learning tasks. We employ a learning rate of $5 \times 10^{-5}$ for the DDPMs or $1 \times 10^{-5}$ LDMs for approximately 300 iterations with a batch size of 40 on $\times 8$ NVIDIA A100.

**Measurements.** In our evaluation of generation diversity, we utilize Intra-LPIPS and FID as described in CDC (Ojha et al., 2021). For Intra-LPIPS, we generate 1,000 images, which each of them wiil be assigned to the training sample with the smallest LPIPS distance. The Intra-LPIPS measurement is obtained by averaging the pairwise LPIPS distances within the same cluster and then averaging these results across all clusters. FID is a widely use metric for assessing the generation quality of generative models by calculating the distribution distances between generated samples and datasets. However, FID may become unstable and unreliable when applied to datasets with few

Table 1: The Intra-LPIPS (↑) results for both DDPM-based strategies and GAN-based baselines are presented for 10-shot image generation tasks. These tasks involve adapting from the source domains of FFHQ and LSUN Church. The "Parameter Rate" column provides information regarding the proportion of parameters fine-tuned in comparison to the pre-trained model's parameters. The best results are marked as **bold**.

| Methods | Parameter Rate | FFHQ → Babies | FFHQ → Sunglasses | FFHQ → Raphael's paintings | LSUN Church → Haunted houses | LSUN Church → Landscape drawings |
|---|---|---|---|---|---|---|
| TGAN | 100% | 0.510±0.026 | 0.550±0.021 | 0.533±0.023 | 0.585±0.007 | 0.601±0.030 |
| TGAN+ADA | 100% | 0.546±0.033 | 0.571±0.034 | 0.546±0.037 | 0.615±0.018 | 0.643±0.060 |
| EWC | 100% | 0.560±0.019 | 0.550±0.014 | 0.541±0.023 | 0.579±0.035 | 0.596±0.052 |
| CDC | 100% | 0.583±0.014 | 0.581±0.011 | 0.564±0.010 | 0.620±0.029 | 0.674±0.024 |
| DCL | 100% | 0.579±0.018 | 0.574±0.007 | 0.558±0.033 | 0.616±0.043 | 0.626±0.021 |
| DDPM-PA | 100% | 0.599±0.024 | 0.604±0.014 | 0.581±0.041 | 0.628±0.029 | 0.706±0.030 |
| DDPM-TAN (Ours) | 1.3% | 0.592±0.016 | 0.613±0.023 | **0.621**±0.068 | 0.648±0.010 | 0.723±0.020 |
| LMD-TAN (Ours) | 1.6% | **0.601**±0.018 | **0.613**±0.011 | 0.592±0.048 | **0.653**±0.010 | **0.738**±0.026 |

samples, such as the 10-shot datasets used in this paper. Following the DDPM-PA approach, we provide FID evaluations using larger target datasets, such as Sunglasses and Babies, which consist of 2,500 and 2,700 images, respectively.

**Baselines.** To adapt pre-trained models to target domains using a limited number of samples, we compare our work with several GAN-based and DDPMs baselines that share similar objectives. These include TGAN (Wang et al., 2018), TGAN+ADA (Karras et al., 2020a), EWC (Li et al., 2020), CDC (Ojha et al., 2021), DCL (Zhao et al., 2022), and DDPM-PA (Zhu et al., 2022). All these methods are implemented on the same StyleGAN2 (Karras et al., 2020b) codebase.

## 5.3 Overall Performance

**Qualitative Evaluation.** Figure 3 presents samples from GAN-based and DDPM-based methods for 10-shot LSUN Church → Landscape drawings (top) and FFHQ → Raphael's paintings (bottom). The samples generated by GAN-based baselines contain unnatural blurs and artifacts. This illustrates the effectiveness of our approach in handling complex transformations while maintaining the integrity of the original image features. Whereas the current DDPM-based method, DDPM-PA (third row), seems to underfit the target domain images, resulting in a significant difference in color and style between the generated images and the target images. Our method preserves many source domain shapes and outlines while learning more about the target style. As demonstrated in Figure 1, our method, TAN, maintains more details such as buildings (above) and human faces (below) in the generated images. Moreover, TAN-generated images exhibit a color style closer to the target domain, especially when compared with DDPM-PA. Compared to other methods, our approach (based on both DDPMs and LDMs) produces more diverse and realistic samples that contain richer details than existing techniques.

**Quantitative Evaluation.** In Table 1, we display the Intra-LPIPS results for DPMs-TAN under various 10-shot adaptation conditions. DDPM-TAN yields a considerable improvement in Intra-LPIPS across most tasks when compared with other GAN-based and DDPMs-based methods. Furthermore, LMD-TAN excels be-

Table 2: FID (↓) results of each method on 10-shot FFHQ → Babies and Sunglasses. The best results are marked as **bold**.

| Methods | TGAN | ADA | EWC | CDC | DCL | PA | ADMT |
|---|---|---|---|---|---|---|---|
| Babies | 104.79 | 102.58 | 87.41 | 74.39 | 52.56 | 48.92 | **46.70** |
| Sunglasses | 55.61 | 53.64 | 59.73 | 42.13 | 38.01 | 34.75 | **20.06** |

yond state-of-the-art GAN-based approaches, demonstrating its potent capability to preserve diversity in few-shot image generation. The FID results are presented in Table 2, where TAN also demonstrates remarkable advancements compared to other GAN-based or DPMs-based methods, especially in FFHQ → 10-shot Sunglasses with 20.06 FID. We provide more results for other adaptation scenarios in the Appendix. Our method can transfer the model from the source to the target domain not only effectively but also efficiently. Compared to other methods that require around 5,000 iterations, our approach only necessitates approximately 300 iterations with limited parameter fine-tuning. The use of TAN enhances efficiency because our approach effectively equates to 3,300 training iterations, marking a notable decrease from the standard 5,000 iterations and DDPM-PA 6,000 - 10,000 equivalent iterations. The time cost of the baseline with adaptor and 5,000 iterations (same with DDPM-PA) is about 4.2 GPU hours, while our model (DPMs-TAN) with only 300 iterations takes just 3 GPU hours.

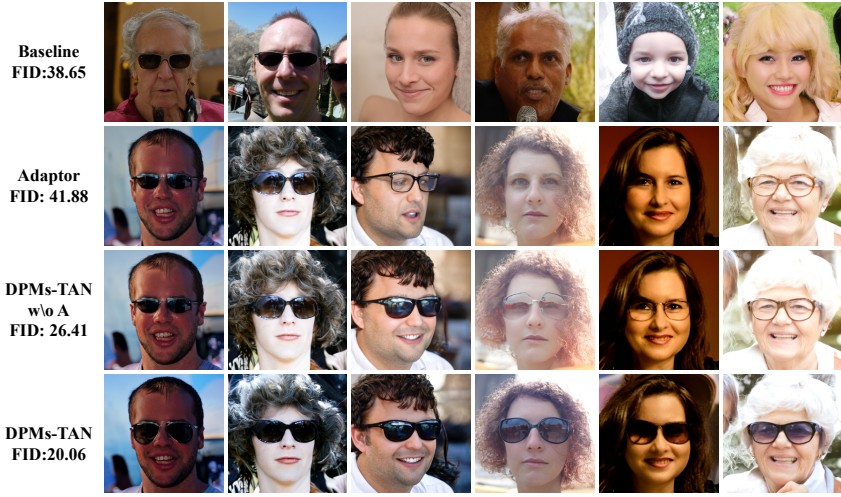

Figure 4: This figure shows our ablation study with all models trained for 300 iterations on a 10-shot sunglasses dataset measured with FID ($\downarrow$): the first line - baseline (direct fine-tuning model), second line - Adaptor (fine-tuning only few extra parameters), third line - DPMs-TAN w/o A (only using similarity-guided training), and final line - DPMs-TAN (our method).

## 5.4 ABLATION ANALYSIS

Figure 4 presents an ablation study, with all images synthesized from the same noise. When compared to directly fine-tuning the entire model (1st row), only fine-tuning the adaptor layer (2nd row) can achieve competitive FID results (38.65 vs. 41.88). The DPMs-TAN without adversarial noise selection (DPMs-TAN w/o A) and all DPMs-TAN (3rd and 4th row) are trained with an extra adaptor layer to save time and GPU memory, and our analysis focuses on the last three rows.

The first two columns demonstrate that all methods can successfully transfer the model to sunglasses, with the TAN containing richer high-frequency details about sunglasses and background items. The 3rd and 4th columns show that the similarity-guided method (3rd row) can produce images of people wearing sunglasses, while the traditional method (2nd row) does not achieve this. The last two columns highlight the effectiveness of the adaptive noise selection method in TAN. The step-by-step transformation showcased in the 5th column provides a clear demonstration of how our method transfers the source face through an intermediate phase, in which the face is adorned with glasses, to a final result where the face is wearing sunglasses. This vividly illustrates the effectiveness of our proposed strategies in progressively boosting the transfer process. The Frechet Inception Distance (FID) scores further illustrate the effectiveness of our proposed strategies; it decreases from 41.88 (with direct adaptation) to 26.41 (with similarity-guided training) and then to 20.66 (with DPMs-TAN), indicating a progressive improvement in the quality of generated images.

Table 3 is the FID and Intra-LPIPS results for classifiers trained on 10 and 100 images. This indicates that only 10 images are sufficient to guide the training process. This effectiveness is largely attributed to the classifiers being trained on noised targeted images among T (1000 steps), ensuring a robust gradient for training.

Table 3: FID and Intra-LPIPS results for classifiers trained on 10 and 100 images.

|  | Intra-LPIPS ($\uparrow$) | FID ($\downarrow$) |
| --- | --- | --- |
| 10-shot classifier | $0.621 \pm 0.068$ | 20.06 |
| 100-shot classifier | $0.637 \pm 0.013$ | 22.84 |

## 6 CONCLUSION

In conclusion, the application of previous GAN-based techniques to DPMs encounters substantial challenges due to the distinct training processes of these models. To overcome this, we introduce TAN to train the DPMs with a novel adversarial noise selection and the similarity-guided strategy that improves the efficiency of the transfer learning process. Our proposed method accelerates training, achieves quicker convergence, and produces images that fit the target style while resembling the source images. Experimental results on few-shot image generation tasks demonstrate that our method surpasses existing state-of-the-art GAN-based and DDPM-based methods, delivering superior image quality and diversity.

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
