# OpenReview forum: "Efficient Transfer Learning in Diffusion Models via Adversarial Noise"
_ICLR.cc/2024/Conference — Submitted to ICLR 2024_

### Official Review · Reviewer_dyX4 · 2023-10-23

**Soundness:** 2 fair
**Presentation:** 2 fair
**Contribution:** 1 poor
**Rating:** 3
**Confidence:** 5

**Summary:**

In this paper, authors propose a novel DPMs-based transfer learning method, TAN, to address the limited data problem. It includes two strategies: similarity-guided training, which boosts transfer with a classifier, and adversarial noise selection which adaptive chooses targeted noise based on the input image. As illustrated in the paper, authors think they haved achieved SOTA results compared with prior works.

**Strengths:**

1.The paper introduces a binary classifier to guide the training and an adversarial way to generate noises, this idea is interesting.
2.The exploration of overfitting in Fig.1 is helpful for the few-shot learning of generative models.

**Weaknesses:**

1.The quantitative results reported for different methods have large overlap when factoring in the std values, in general, the boldfacing of average values when ignoring the stds is not the best practice.
2. I have read several related works and find almost all of them show use samples transferred from FFHQ as qualititative results. I found that the results of this paper on FFHQ are only shown in Supp. I cannot figure out the improvement of this method compared with DDPM-PA on those results. Actually, I think DDPM-PA shows better results.
3. Results in this paper should be compared with more modern text-to-image methods based on diffusion models, including textual inversion, dreambooth, domainstudio. If this method is only applied to traditional methods, it's not convincing enough.
4. In LSUN Church --> Landscape drawings, it seems that DDPM-PA carries out a style transfer process, this work fails to get samples of church actually. Therefore, I wonder if this comparison is fair. For FFHQ --> babies and sunglasses, this work and DDPM-PA share the same target. However, I think DDPM-PA performs better.

**Questions:**

My main concern is about the performance and applicable scenarios. See the weakness part.

---

> ### Author Response · Authors · 2023-11-17
>
> Thank you for your thoughtful review and feedback.
>
> ## 1. **Overlap of Results**
> - **Overlap**: The phenomenon of overlapping results, as suggested by the standard deviation values, is a widespread challenge in this area of study, not just specific to our research. This issue is also evident in methodologies such as CDC, DCL, and DDPM-PA. For example, the Intra-LPIPS result of DDPM-PA on FFHQ $\to$ Babies is 0.599 $\pm$ 0.024 and LSUN Church $\to$ Haunted houses is 0.628 $\pm$ 0.029, while the Intra-LPIPS result of CDC on FFHQ $\to$ Babies is 0.583 $\pm$ 0.014 and LSUN Church $\to$ Haunted houses is 0.620 $\pm$ 0.029.
> - **P Value**: We also present a new table about p-value between our method and DDPM-PA to demonstrate the tangible improvement of our approach. The Inter-LPIPS score, our primary metric, often faces an overlap problem due to the highly competitive advancements in this field. Nevertheless, the p-value table still illustrate significant improvements of our method despite these constraints.
> |  Datasets | Babies | Sunglasses | Raphael's paintings | Haunted houses | Landscape
> |-----|:-----:|:-----:|:-----:|:-----:|:-----:|
> | P-value (AP with TAN) | $2.6 \times 10^{-6}$ | $1.9 \times 10^{-25}$  | $7.0 \times 10^{-54}$  | $8.7 \times 10^{-86}$  | $9.5 \times 10^{-48}$  |
>
> ## 2. **Qualitative Results on FFHQ**
> - In Figure 3 of the main paper, we have already presented samples transferred from the FFHQ dataset, providing a qualitative comparison with DDPM-PA and other methods. The FFHQ $\to$ Raphael's paintings transformation showcases the strengths of our method, particularly in managing complex features. Our approach achieves more accurate color reproduction and fewer artifacts compared to DDPM-PA. We would greatly appreciate it if the reviewer could share any relevant references, enabling us to thoroughly examine and pinpoint any potential discrepancies or overlooked aspects in our approach.
>
>
> ## 3. **Comparison with Modern Text-to-Image Methods**
> - **Different Tasks**: The tasks of text-to-image transfer learning and few-shot image generation are fundamentally different in their objectives and methodologies.
> Text-to-image transfer learning primarily focuses on subject-specific generation.
> It aims to capture the essential features of a target domain while maintaining diversity, even with a limited set of images.
> Renowned methods in this field, such as CDC, DCL, and DDPM-PA, have not typically been applied to text-to-image conditional generation tasks.
> For instance, Dreambooth fine-tunes an existing text-to-image model using a small set of subject-specific images, enabling the model to form a distinct association with that subject.
> In contrast, few-shot image generation concentrates on preserving similarities and differences among instances in the source domain, without a specific focus on maintaining the primary subject of the target images.
>
> - **Dreambooth Results**: Our new experiment, illustrated in Appendix Figure 7, reveals the difficulties Dreambooth faces in few-shot generation tasks, as the generated images closely resemble the target images.
> Additionally, it's challenging to conduct a fair comparison between our method and text-to-image approaches.
> These text-to-image methods typically require pre-training on large text and image paired datasets, enabling them to generate images of the target domain without fine-tuning.
> For example, the model used in Dreambooth can create images of people wearing sunglasses without any fine-tuning.
> The goal of Dreambooth's fine-tuning is to produce images that resemble the training data but set against different backgrounds.

---

> > ### Author Response · Authors · 2023-11-17
> >
> > ## 4. **Fairness in Comparative Analysis**
> > - **LSUN Church to Landscape Drawings**: We question the success of DDPM-PA in the transfer learning task for LSUN Church $\to$ Landscape Drawings. Despite their claim, "We demonstrate the effectiveness of DDPM-PA both qualitatively and quantitatively across several few-shot image generation tasks, showing that DDPM-PA surpasses the generation quality and diversity of current leading GAN-based methods," they do not adhere to the principle of few-shot transfer learning in the target domain. For instance, they are unable to replicate the watercolor brush strokes characteristic of the target images in LSUN Church $\to$ Landscape Drawings tasks. We have followed the same few-shot image generation protocol as CDC and DDPM-PA in our experiments, involving fine-tuning a pre-trained image generation model for the target domain. Since DDPM-PA used the same baselines as CDC and DCL for their visual results, we believe our selection of results in our paper is justified.
> >
> > - **FFHQ to Babies and Sunglasses**: The images generated by our method are more in line with estahlished methods, underscoring the superiority of our approach. It's worth noting that the sunglasses images produced by DDPM-PA exhibit a distinct bluish hue and increased background artifacts. Furthermore, our method shows a significant improvement in FID scores, achieving 20.06 (TAN) versus 34.75 (PA) in the FFHQ $\to$ Sunglasses task. This objectively indicates that our method performs better.

---

> ### Author Response · Authors · 2023-11-21
> **Additional User Study on Qualitative Performance**
>
> I hope this message finds you well. I am writing to express my gratitude for your thoughtful review and feedback on our work. We have carefully considered your comments and have prepared a detailed rebuttal to address your queries and concerns.
>
> In our rebuttal, we have clarified the quantitative improvement demonstrated by our results, differentiated our approach from current text-to-image methods, and validated the fairness of our comparative analyses with comprehensive statistical and visual evidence.
>
> We carried out an additional anonymous user study to assess the qualitative performance of our method in comparison to DDPM-PA.
> In this study, participants were shown three sets of images from each dataset, featuring DDPM-PA, our method (DDPM+TAN), and images from the target domain.
> For each set, we displayed five images from each method or the target image, as illustrated in our main paper.
> To maintain anonymity and neutrality, we labeled the methods as A/B instead of using the actual method names (PA and TAN).
> We recruited volunteers through an anonymous online platform for this study. During the study, participants were tasked with choosing the set of images (labeled as A or B, corresponding to PA or TAN) that they believed demonstrated higher quality and a closer resemblance to the target image set.
>
> |  Methods  | Sunglasses | Babies | Landscape  | Raphael's paintings | Average
> |-----|:-----:|:-----:|:-----:|:-----:|:-----:|
> | PA | 20.0% | 33.3% | 20.0%  | 33.3%  | 26.65%  |
> | TAN | 80.0% | 66.7% | 80.0%  | 66.7%  | 73.35%  |
>
> Of the 60 participants, a significant 73.35% favored our method (DDPM+TAN), indicating that it produced images of superior quality and more effectively captured the intricate types of target domains. While this experiment did not comprehensively account for factors such as the participant's gender, age, regional background, and others, the results nonetheless suggest that our images possess better visual quality to a notable extent.
>
> We kindly request you to review our rebuttal your feedback is invaluable to us, and we would greatly appreciate your insights on our responses.
>
> Thank you for your time and consideration.

---

### Official Review · Reviewer_ft3z · 2023-10-30

**Soundness:** 3 good
**Presentation:** 3 good
**Contribution:** 3 good
**Rating:** 6
**Confidence:** 3

**Summary:**

The paper presents an innovative approach to transfer learning in diffusion models with sparse target data. To tackle the limited data problem, the authors propose TAN, a new diffusion-based method including two strategies: similarity-guided training, which aims to enhance transfer with a classifier, and adversarial noise selection to improves the efficiency of training process. The authors conducted extensive experiments in the context of few-shot image generation tasks and demonstrated that their method is not only efficient but also excels in terms of image quality and diversity when compared to existing GAN-based and DPM-based methods.

**Strengths:**

1.	This paper is well written, easy to follow and conducts a lot of experiments  with comprehensive analysis, demonstrating the effectiveness of the proposed approach.
2.	The design of the similarity-guided training and adversarial noise is intuitive and reasonable.

**Weaknesses:**

1.	The graphical representation of the results doesn't clearly demonstrate a significant improvement compared to other existing methods.
2.	Regarding the approach to adversarial noise selection for Eq. 7 and Eq. 8, could you provide further clarification on the choice of minimizing the maximum Gaussian noise at step t? Additionally, since optimizing for maximum noise can be particularly challenging, could you delve deeper into how the multi-step variant of PGD with gradient ascent ensures the performance of this methodology? Expanding on these aspects would significantly enhance the comprehensibility and value of this paper.
3.	The paper lacks an in-depth exploration of the implications for time and GPU memory conservation when utilizing the supplementary adaptor module and the process of adversarial noise selection. To enhance the overall quality of the paper, I suggest providing a more comprehensive elucidation, along with relevant graphical representations, regarding the efficient training procedures involving these components.

**Questions:**

My main concerns and questions lie in the weaknesses. The author should discuss them in detail.

---

> ### Author Response · Authors · 2023-11-17
>
> Thank you for your thoughtful review and feedback on our work. We will address each of your queries, clarifying any misunderstandings and delving into pivotal academic questions that merit exploration. We are also pleased to share some of our insights with you.
>
> ### 1. **Qualitative Performance and Comparison with Existing Methods**
> - **Illustration of Performance**: Figure 3 in our study showcases our method's notable advancements over existing approaches, particularly DDPM-PA. Our technique effectively captures the intricate type of target domains. This is clearly visible in the top and bottom rows of the figure, where our method (4th and 5th images) aligns more closely with the target domain's type than DDPM-PA (3rd image).
>
> - **Specific Examples**: We provide a detailed comparison of certain features, such as the watercolor brush strokes in the up images of churches and mountains, and the red hat or yellow curly hair in the bottom row images. These examples further highlight our method's enhanced ability to adapt to the target domain's type, offering a significant improvement over the results achieved by DDPM-PA.
>
> ### 2. **PGD Step Number and Adversarial Noise Selection**
> - **Min Max at Timestep t**: As detailed in Equation 8 of the main paper, we employ a multi-step variant of Projected Gradient Descent (PGD) for the inner maximization. This approach helps in identifying the "worst-case" adversarial noise, which corresponds to the current state of the neural network. Subsequently, we use this adversarial noise as the "maximum" noise input to minimize our model's loss, as outlined in Equation 9 of our paper.
> - **Ensures the Performance**: The choice of maximizing Gaussian noise at step t is strategically aligned with our aim to challenge the model with the “worst-case” scenarios. This approach ensures that the model is robust enough to handle the most difficult noise patterns, thereby improving overall performance. As demonstrated in Section 5.1 of the main paper, the Adervsrial noise with PDG can rectify the issue of incorrect gradients caused by limited data. Figure 2 (a) in main paper shows the output layer gradient direction for four different settings in the first iteration, all with identical noise and timestep $t$.
> The red line, derived from ten thousand varied samples, serves as a dependable reference direction (approximately 45 degrees southwest).
> For comparison, 10-shot samples are repeated a thousand times in one batch, aligning them with the ten thousand distinct samples.
> The dark blue and sienna lines represent the gradient computed using traditional DDPM as the baseline and similarity-guided training in a 10-shot sample, respectively.
> The orange line illustrates our method, DDPM-TAN, applied to a 10-shot sample.
> The gradient with adversarial noise (represented by the orange line) is closer to the reliable reference direction than the gradient without adversarial noise (illustrated by the sienna line). This proximity indicates the method's effectiveness in ensures the prformance.
>
> ### 3. **Time and GPU Memory Conservation**
> - **GPU Memory**: To further elucidate these aspects, we will include graphical representations in the revised version of the paper, showcasing the efficiency gains in training procedures involving these components. The Table in Appendix A.5 illustrates the GPU memory usage for each module in batch size 1, comparing scenarios with and without the use of an adaptor. It reveals that our module results in only a slight increase in GPU memory consumption.
> - **Time**: In Section 5.3 of our main paper, we discuss the time efficiency of our method. Our approach not only effectively but also efficiently transfers the model from the source to the target domain.
> Compared to other methods requiring about 5,000 iterations, our method needs only approximately 300 iterations with limited parameter fine-tuning. The incorporation of Training Adversarial Noise (TAN) further enhances efficiency. Effectively, our approach equates to 3,300 training iterations, significantly less than the standard 5,000 iterations and the 6,000 - 10,000 equivalent iterations needed for DDPM-PA.
> In terms of time cost, the baseline with an adaptor undergoing 5,000 iterations (the same as DDPM-PA) consumes about 4.2 GPU hours. In contrast, our model (DPMs-TAN) requires just 3 GPU hours for its 300 iterations, demonstrating notable time efficiency.

---

> > ### Author Response · Authors · 2023-11-22
> >
> > I hope this message finds you well. I am writing to express my gratitude for your thoughtful review and feedback on our work. We have carefully considered your comments and have prepared a detailed rebuttal to address your concerns.
> >
> >
> > In our rebuttal, we highlight our method's superior qualitative performance compared to existing methods. We detail the effective use of PGD in selecting adversarial noise to enhance model robustness. Additionally, we demonstrate our approach's efficiency in GPU memory utilization and training time. We kindly request you to review our rebuttal and your feedback is invaluable to us, and we would greatly appreciate your insights on our responses.
> >
> > Thank you for your time and consideration.

---

### Official Review · Reviewer_Pd9e · 2023-11-01

**Soundness:** 3 good
**Presentation:** 3 good
**Contribution:** 3 good
**Rating:** 6
**Confidence:** 3

**Summary:**

Generative models can leverage large-scale datasets to learn and produce diverse high-quality outputs. To overcome the limitations of this approach, methods for transfer learning from models trained on extensive datasets have been studied. However, diffusion-based generative models, unlike directly step-wise generative models like GANs, generate samples through the diffusion of noises over a large number of steps, making it challenging to apply conventional transfer learning methods. Therefore, in this paper, a method called TAN, which utilizes adversarial noises to effectively transfer learning to diffusion models.

Generating high-quality and diverse results with generative models comes with the challenge of obtaining a large amount of training data and stabilizing the training process. In this regard, transfer learning and few-shot learning research are important, yet relatively less explored. Previous studies attempted transfer learning by matching the results of intermediate steps of DDPM, but they led to distorted transfer and overfitting issues. the proposed approach, incorporating similarity-guided training and adversarial noise selection techniques, yielded well-transferred results.

**Strengths:**

This paper proposes simple yet effective methods, namely similarity-guided training and adversarial noise, for transfer learning with diffusion models.

The proposed methods yielded high-quality results and were demonstrated through various experiments.

The paper presents equations 4 and 5, which induce the Kullback-Leibler divergence between the source and target models during the reverse process of the diffusion model.

This divergence is defined as similarity and utilized to control transfer learning.

The proposed similarity-guided approach by the authors not only induces overall transfer for the target domain but also considers the characteristics of the diffusion generative model.

By utilizing adversarial noise selection, the method allows the noise to fit more accurately, resulting in high-quality output and addressing the issues faced by previous methods.

The paper presents a cohesive and easily understandable writing flow, effectively conveying the underlying arguments and the rationale of the study.

**Weaknesses:**

I think it would be better if there is deeper analysis regarding the similarity or adversarial noise in the proposed method. It would be even better if there were various experiments and analyses to examine the semantic effects of the redefined reverse step compared to the vanilla model. I am curious about the author's insights beyond mathematically deriving the KL divergence between the source and target domains, exploring different aspects.

**Questions:**

1. I am curious if this method can be applied not only to transfer learning or few-shot learning but also to fields such as domain adaptation or domain control.

2. I wonder if the adversarial noise in this method can assist in achieving higher-quality results in the training of general DDPMs.

3. I'm curious if this method has been experimented with in domains other than 2D images.

---

> ### Author Response · Authors · 2023-11-17
>
> Thank you for your thoughtful review and feedback on our work. We will address each of your queries, clarifying any misunderstandings and delving into pivotal academic questions that merit exploration. We are also pleased to share some of our insights with you.
>
> ### 1. **Additional Insights and Analysis**
> - **Further Analysis and Visualization**: In response to your request for deeper analysis, we provide a new experiment in Appendix A.4 about generated during each iteratuions as our exploration of similarity and adversarial noise.
> This include more detailed visualizations, especially comparing the reverse image of our method with that of the standard model with the different training iteraion step.
> As shown in Figure 5, the first row demonstrate that the orangial train the DPMs with limited iterations is hard to get a successfully transfer. The second raw shows that training with our similarity-guide method can boost the convergence to the taget domain. The third rows shows that training further with adversrial noise can even more faster converge. As shown the 150 iteration of right pictures, compared with the training only with similarity-guide (2nd row) TAN can get the face with sunglasses image.
> <!-- This expanded analysis aims to provide more comprehensive insights that our method can boost the transfer learning. -->
> - **KL Divergence Analysis**: Our methodology employs KL divergence between the source and target domains as a fundamental analytical tool.
> This approach aids in understanding the differences between well-established source and target models and also provides insights into effectively utilizing similarities in transfer learning for DPMs.
> We initially leverage existing knowledge by fine-tuning a diffusion model derived from a pre-trained model.
> Therefore, examining the differences between the pre-trained model in the source domain and the fine-tuned model in the target domain is essential
> Furthurmore, equation 5 in our paper, $$\texttt{D}_{\textrm{KL}} \left( p_{\theta_{\textrm{S}}, \phi}(x_{t-1}^{\textrm{S}} | x_{t}), p_{\theta_{\textrm{T}}, \phi}(x_{t-1}^{\textrm{T}} | x_{t}) \right)
> = \mathbb{E}_{t, x_0, \epsilon} \left [ \left\| \nabla_{x_t} \log p_{\phi}(y=\textrm{S}|x_t) - \nabla_{x_t} \log p_{\phi}(y=\textrm{T}|x_t) \right\|^2 \right ]$$, demonstrates how the divergence depends on $x_t$ at specific timesteps $t$.
> This highlights that transfer learning in DPMs can efficiently leverage this divergence for more effective training.
>
> ### 2. **Application Beyond Transfer and Few-shot Learning**
> - **Domain Adaptation, Control and Other Domains**: Our method, centered on adversarial noise and similarity-guided training, can indeed be extended beyond transfer and few-shot learning.
> It holds potential in domain adaptation and control tasks, especially where the target domain has limited data but is somewhat related to a well-trained source domain.
> Our current focus has been predominantly on 2D images.
> However, the underlying principles of our method are adaptable to other domains.
> Thank you for your suggestion. We intend to explore the application of our technique across various fields and domains, including domain adaptation and 3D modeling, among others. This will enable us to fully utilize the versatility of our approach in our future work.
>
> ### 3. **Adversarial Noise in General DPMs Training**
> Our method is applicable in general DPMs training, but the effectiveness of adversarial noise might be reduced during the initial unstable gradient phase and when dealing with large datasets.
>
> - **Training From Scratch**: PGD may not be effective in the initial stages of the training process.
> The model's gradient is typically unstable when training begins from scratch, and adversarial noise requires a relatively accurate model gradient to function effectively.
> If the initial model's gradient is unstable, it fails to provide beneficial adversarial noise for training.
>
> - **Large Data Sets**: The application of adversarial noise has substantially enhanced the training of DPMs under conditions of limited data and iterations.
> As shown in Figure 2 (Page 6) of our paper, the adversarial noise method helps address incorrect gradients, which is advantageous for training on smaller datasets.
> For larger datasets, however, the benefits of our approach might be less noticeable.
> This is due to the extensive iterations required for model convergence, as represented by the cyan line (associated with 10,000 points data) in Figure 2.
>
> Therefore, although adversarial noise shows promising performance in few-shot transfer learning, its effectiveness may be less pronounced in general DPMs training.

---

> ### Author Response · Authors · 2023-11-22
>
> I hope this message finds you well. I am writing to express my gratitude for your thoughtful review and feedback on our work. We have carefully considered your comments and have prepared a detailed rebuttal to address your concerns.
>
> In our rebuttal, we clarify our methodology's nuances with added experiments and visualizations. Our discussion extends to the method's applicability beyond transfer and few-shot learning. Additionally, we explore the impact of adversarial noise in training general DPMs, considering various training scenarios and dataset sizes. We kindly request you to review our rebuttal and your feedback is invaluable to us, and we would greatly appreciate your insights on our responses.
>
> Thank you for your time and consideration.

---

> > ### Comment · Reviewer_Pd9e · 2023-11-23
> >
> > Thanks for your comments for my opinion. The additional experiments and also analysis can be helpful to my understaning. so I would like to keep my rating.

---

### Official Review · Reviewer_jP4S · 2023-11-01

**Soundness:** 3 good
**Presentation:** 3 good
**Contribution:** 3 good
**Rating:** 6
**Confidence:** 2

**Summary:**

This paper proposes a Diffusion Probabilistic Model (DPM)-based transfer learning method, i.e., to address the limited data problem. Specifically, similarity-guided strategy is designed to enhance the few-shot transfer learning and an adversarial noise selection method is proposed to address the underfitting problem during a limited number of iterations. Both qualitative and quantitative results demonstrate the effectiveness of the proposed method.

**Strengths:**

The paper is well-written with clear description.

It analyzes the limitations of DPM-based transfer learning and presents reasonable solutions.

**Weaknesses:**

How does the pre-trained DPMs come and if the proposed method can be applied on other pre-trained DPMs for transfer learning?

If the authors could provide visualizations on the selected noise for deeper analysis on its effect?

The step number used to perform the projected gradient descent (PGD) in Eq.(7)?

**Questions:**

Please refer to the Weaknesses for details.

---

> ### Author Response · Authors · 2023-11-17
>
> Thank you for your thoughtful review and feedback on our work. We will address each of your queries, clarifying any misunderstandings and delving into pivotal academic questions that merit exploration. We are also pleased to share some of our insights with you.
>
> ### 1. **Pre-trained DPMs**
> - **Clarification on Pre-trained DPMs**: We provide additional details in Section 5.2 (Experimental Setup) about the pre-trained DDPM (Denoising Diffusion Probabilistic Models)[1] and LDM (Latent Diffusion Models)[2]. The pre-trained DDPM are from [1] and pre-trained LDM are from [2].
> - **Generalization of Our Method**: Our method is general on different DPMs' pre-trained model since our method focuses on the how to transfer based on DPMs' theories instead of the structure of neural networks.
> Our method's generalizability is demonstrated through its successful application to two well-known DPM pipelines, DDPM and LDM. This highlights the method's versatility and wide applicability in various transfer learning scenarios within the DPM framework.
>
> ### 2. **Visualizations of Adversarial Noise**
> - **2D Visualization**: The background red point of Figure 2 (a) (page 6) presents a 2D visualization of adversarial noise, showing a transition from a standard Gaussian distribution (circular shape) to an elliptical one. The elliptical shape aligns with the gradient direction of the model's parameters. This illustrates how our approach fine-tunes the model by focusing on the most challenging noise scenarios.
> - **Challenges in Visualizing High-Dimensional Adversarial Noise**: Visualizing high-dimensional adversarial noise, especially for real images involving vectors of 3 * 256 * 256 dimensions, is significantly challenging. These visualizations can be less intuitive and may struggle to convey critical information due to their complexity and high dimensionality.
>
> ### 3. **PGD Step Number in Adversarial Noise Selection**
> - **PGD Step Number**: In all experiments presented in our paper, we set the PGD step number to \( J = 10 \) to demonstrate the generalizability of our method, as detailed in subsection 5.2 of the main paper.
>
> [1] Ho, J., Jain, A. and Abbeel, P., 2020. Denoising diffusion probabilistic models. Advances in neural information processing systems, 33, pp.6840-6851.
>
> [2] Rombach, R., Blattmann, A., Lorenz, D., Esser, P. and Ommer, B., 2021. High-resolution image synthesis with latent diffusion models. 2022 IEEE. In CVF Conference on Computer Vision and Pattern Recognition (CVPR) (pp. 10674-10685).

---

> ### Author Response · Authors · 2023-11-22
>
> I hope this message finds you well. I am writing to express my gratitude for your thoughtful review and feedback on our work. We have carefully considered your comments and have prepared a detailed rebuttal to address your concerns.
>
> In our rebuttal, we clarify the origins and features of the pre-trained DPMs utilized in our study. We also provide details on our visualization techniques for adversarial noise. Additionally, we specify the PGD step number used in our experiments, highlighting the robustness and versatility of our approach. We kindly request you to review our rebuttal and your feedback is invaluable to us, and we would greatly appreciate your insights on our responses.
>
> Thank you for your time and consideration.

---

### Author Response · Authors · 2023-11-17

Thank you for your thoughtful review and feedback on our work. We will address each of your queries sequentially, clarifying any misunderstandings and delving into pivotal academic questions that merit exploration. We are also pleased to share some of our insights with you.


---

In response to the reviews, we have included additional experiments in the Appendix, marked in red for easy identification:
1. **Quantitative Evaluation of Different Iterations**: This is an added analysis, focusing on each module within our method to Reviewer [Pd9e].
2. **GPU Memory**: This section details the GPU memory consumption for each module in our method to Reviewer [ft3z].
3. **Dreambooth**: We have included Dreambooth-generated images for the FFHQ $\to$ Sunglasses scenario in response to Reviewer [dyX4].

---

### Author Response · Authors · 2023-11-23
**Rebuttal Due**

Dear Reviewers,

We sincerely appreciate the time and effort you have dedicated to reviewing our work. As the deadline for our discussion period approaches, we kindly request that you inform us of any remaining queries. We have addressed the comments and concerns raised during the review process and have provided detailed responses to your concerns.

We understand the demands of your schedules and are deeply grateful for your commitment to reviewing our work. We eagerly await your feedback and hope to continue this productive dialogue.

Thank you for your consideration.

Warm regards,

---

### Meta-Review · Area_Chair_XKJQ · 2023-12-06

**Metareview:**

The paper addresses the problem of taking a large pre-trained generative model and applying or adapting it to a target distribution which has a small number of data points. The paper proposes a similarity-guided training which uses a classifier to estimate the difference between the source and the target domains, which stabilizes the training, and a adversarial noise selection to dynamically choose the noise according to the input image -- this minimizes the "worse-case" noise, speeding up the training. The paper evaluates on several transfer tasks: FFHQ->Babies, FFHQ->Sunglasses, FFHQ->Raphael's paintings, LSUN Church->Haunted House, LSUN Church->Landscape drawings and in the  suppl, FFHQ->Sketches, FFHQ->Amedeo's paintings. The paper shows a very slightly improvement on the Intra-LPIPs for 5 of these tasks and show a qualitative comparison which is not very convincing. The main thing that would strengthen the paper are more convincing qualitative and quantitative results or moving towards more modern techniques.

**Justification For Why Not Higher Score:**

I think this paper could be accepted but currently the results seem a bit weak both quantitatively and qualitatively and I'm sympathetic to the point that this is a really niche area using relatively older techniques. Improving the quality of the outputs or upgrading the techniques to be more modern would substantially help the paper.

**Justification For Why Not Lower Score:**

N/A

---

### Decision · Program_Chairs · 2024-01-16

Reject